# Establishing standardized conditions for clinically available sound-localization tests: A multicenter approach

Tadao Yoshida[1]*, Takashi Ishino[2], Satoshi Iwasaki[3], Naoki Oishi[4], Yusuke Matsuda[5], Tetsuya Tono[5], Kazuma Sugahara[6], Hiroshi Yamazaki[7], Sumito Jitsukawa[8], Hiroshi Nakanishi[9], Ryosuke Kitoh[10], Takashi Sato[11], Masumi Kobayashi[1], Kazuki Nishida[12], Takeshi Nakaichi[13]

1 Department of Otorhinolaryngology, Nagoya University Graduate School of Medicine, Nagoya, Japan, 2 Department of Otorhinolaryngology, Head and Neck Surgery, Graduate School of Biomedical Sciences, Hiroshima University, Hiroshima, Japan, 3 Department of Otorhinolaryngology, International University of Health and Welfare Mita Hospital, Tokyo, Japan, 4 Department of Otorhinolaryngology, Head and Neck Surgery, School of Medicine, Keio University, Tokyo, Japan, 5 Department of Otorhinolaryngology, International University of Health and Welfare Hospital, Nasu, Tochigi, Japan, 6 Department of Otolaryngology, Yamaguchi University Graduate School of Medicine, Ube, Yamaguchi, Japan, 7 Department of Otolaryngology, Head and Neck Surgery, Graduate School of Medicine, Kyoto University, Kyoto, Japan, 8 Department of Otolaryngology, Head and Neck Surgery, Sapporo Medical University School of Medicine, Sapporo, Japan, 9 Department of Otorhinolaryngology, Head and Neck Surgery, Hamamatsu University School of Medicine, Hamamatsu, Japan, 10 Department of Otolaryngology, Shinshu University School of Medicine, Nagano, Japan, 11 Department of Otorhinolaryngology–Head and Neck Surgery, Graduate School of Medicine, Osaka University, Osaka, Japan, 12 Department of Biostatistics Section, Center for Advanced Medicine and Clinical Research, Nagoya University Graduate School of Medicine, Nagoya, Japan, 13 Research and Development Center, RION Co. Ltd, Tokyo, Japan

* yoshida.tadao.h6@f.mail.nagoya-u.ac.jp

## Abstract

Sound localization is essential for auditory spatial awareness. The process relies on interaural differences in timing and level, and spectral cues. This study aimed to standardize sound-localization testing conditions across facilities in Japan, analyze the impact of early reflected sounds on localization accuracy, and compare outcomes between individuals with normal hearing and those with unilateral hearing loss. This study included 77 participants with normal hearing and 45 individuals with unilateral hearing loss, at 11 facilities. Sound-localization tests were conducted using nine loudspeakers arranged in a 180° horizontal arc. The stimuli consisted of Comité Consultatif International Téléphonique et Télégraphique (CCITT) and low-pass CCITT noise bursts at randomized levels of 50, 55, and 60 dB SPL. The reflected sound measurements employed time-stretched pulses to analyze early reflections (4–7 ms). The localization accuracy was assessed using the root-mean-square error and mean deviation score. Localization performance was negatively influenced by early reflections, with reflected sound envelope area and peak values within 4–7 ms correlating significantly with reduced accuracy (r = −0.535 to −0.555). Participants with normal hearing achieved a root-mean-square error of 2.0° ± 4.8°, whereas participants with

**Data availability statement:** All relevant data are within the manuscript and its Supporting Information files.

**Funding:** This study was supported by the Japan Agency for Medical Research and Development (AMED) under Grant Number JP24dk0310128. The funders had no role in study design, data collection and analysis, decision to publish, or preparation of the manuscript.

**Competing interests:** I have read the journal's policy and the authors of this manuscript have the following competing interests: This was a joint research project with RION Co., Ltd., which assisted purely in measuring the reflected sound. RION Co., Ltd. was not involved in any other aspect of this research, including the planning and implementation, data collection, analysis and interpretation, decision to publish, preparation, review, or approval of the paper.

unilateral hearing loss exhibited significantly greater errors (68.4°±40.7°, p<.001). Asymmetries in the left–right response accuracy correlated positively with the reflected sound characteristics (r>0.6). Noise type (normal vs. low-pass CCITT) did not significantly impact performance in either group. Early reflections significantly compromise sound-localization accuracy, particularly in smaller testing environments where reflections overlap with direct sounds. Standardized testing protocols, in which early reflections are controlled, are critical for reliable assessments. The use of sound-absorbing materials can enhance the test precision, particularly in the clinical evaluation of unilateral hearing loss. These findings emphasize the need for optimizing acoustic conditions to improve the reliability and accuracy of sound-localization testing.

## Introduction

Sound localization is the identification of the location of a sound source based on auditory input. In other words, it involves the determination of the direction of an object to be sought or avoided, or the direction in which attention should be directed, which is an essential ability for animals, including humans.

Two directions are involved in the perception of directionality, i.e., horizontal and vertical, and sound information is required for each perception. In the horizontal direction, the interaural time difference (ITD), which is the difference in time between sounds entering the left and right ears, and the interaural level difference (ILD), which is the difference in sound pressure between the left and right ears, provide important sound information. ITDs and ILDs are mainly used to localize sounds with frequencies below and above 1500 Hz, respectively [1]. The interaural phase difference (IPD) is an important auditory spatial cue that becomes particularly useful when the ITD and ILD are not sufficiently reliable. The IPD refers to the difference in the phase of a sound wave arriving at each ear, and it is especially effective at lower frequencies where phase information is more discernible and ITD and ILD cues can be ambiguous or diminished [2,3].

On the other hand, even though ITDs and ILDs cannot be utilized in the vertical midplane, people with normal hearing can perceive direction as they can utilize the increase or decrease in frequency caused by changes in the direction of the sound source. When people hear a sound, the spectral component changes are caused by the reverberation effect of the auricle and the head-shadow effect. Because a change in the spectral component depends on the direction of the sound source, directional perception can be achieved by detecting these changes. This change in sound frequency that can be used for directional perception is called a spectral cue, which is the sound information needed not only for vertical perception but also for horizontal perception [4,5]. Thus, since these four elements (ITD, ILD, IPD, and spectral cues) are important in directional perception, a test method that reflects these detection capabilities is needed for accurate sound-localization testing.

In general, sound-localization test results are worse in patients with unilateral hearing loss (UHL) than in healthy participants, and directional perception (i.e., the ability to recognize the direction of a sound source) on the side of the hearing-impaired ear is worse than that on the side of the normal-hearing ear. This result has been attributed primarily to ITD and ILD, which can be used by individuals with binaural hearing, but are often substantially reduced or unreliable in individuals with UHL owing to limited or degraded input from the affected ear [6,7].

In this study, we aimed to establish an optimized and standardized sound-localization test by analyzing the results of normal-hearing control and UHL cases in facility-specific environments by using standardized testing equipment (S1 Fig).

## Materials and methods

### Participants

This study was conducted from August 2023 to September 2024. Seventy-seven patients, including seven volunteers from each of the eleven university hospitals involved in the study, were included in a normal-hearing control group (mean age: 36.5 years; range, 20–68 years). The control group comprised individuals with an average hearing level of 25 dB HL or less at four frequencies (500, 1000, 2000, and 4000 Hz) in both the left and right ears based on pure-tone hearing levels. In addition, 45 patients with UHL, defined as an average hearing level of 70 dB HL or greater in the worse-hearing ear and 40 dB HL or less in the better-hearing ear, were included in the study. The mean age of the 45 patients with UHL was 57.4 years (20–75 years). In cases of mixed hearing loss, patients with a mean bone-conducted hearing level of 55 dB or greater in the worse-hearing ear were included in the study. Patients with hearing fluctuations were excluded.

### Test environment

Sound-localization tests were conducted in various environments at 11 facilities. Room sizes, floors, ceilings, wall materials, and door materials were reported for all tested environments. One facility used an anechoic room and the other 10 used soundproof or semi-soundproof rooms. The room sizes ranged from 240 to 453 cm (width), 211–327 cm (height), and 207–512 cm (depth). In the anechoic room, all room surfaces were sound absorbing, and no windows were present. In the remaining facilities, nine used carpets as flooring material and one used wooden flooring. Only four facilities had sound-absorbing materials on their ceilings. Eight facilities had sound-absorbing materials on their sides and two had concrete sides. Sound-absorbing sponges were attached to the concrete walls and metal parts as a countermeasure against the reflected sound. Five facilities reported having metallic door sections. Background noise was below 30 dBA at all sites.

### Measurement of reflected sound

Sounds were presented as time-stretched pulses (TSP), which are commonly used in impulse-response measurements in acoustic rooms. The TSP is a sweep sound that can be used to calculate the impulse response in an acoustic room by convolving the reverse TSP signal with the recorded signal. For the measurement of the impulse response in the test room in this experiment, TSPs were presented at 70 dB SPL of 0.5-s duration, using a loudspeaker (SRS-XB01, Sony, Tokyo, Japan) from positions at 100 cm, from channel (ch) 1–9 (Fig 1). They were recorded three times in the participant's position (110 cm from the floor). For each loudspeaker position (channels 1–9), TSPs were presented three times, and the recording with the least background noise among the three trials was selected for the evaluation of the impulse response. A sound-level meter (NA-28; Rion, Tokyo, Japan) was used for the measurements. Thus, the measured impulse-response duration at 21 ms had a direct sound arising from approximately 3 ms (approximately 100 cm from the speaker to the sound-level meter microphone), and its peak value was normalized as 1 and included reverberation and reflected components that appeared as fluctuations within the waveform envelope. For each facility, the response time was calculated considering the room size and the margin of error of ± 5 cm for both the sound-level meter and the reflected sound measured from the floor and ceiling, with a corresponding time range identified within the envelopes. An example of the obtained acoustic waveform is shown in Fig 2.

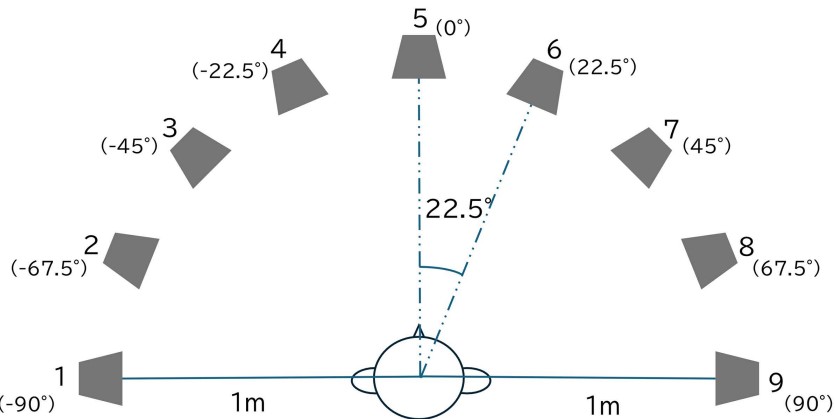

**Fig 1. Participant and loudspeaker placement for sound-localization tests.**

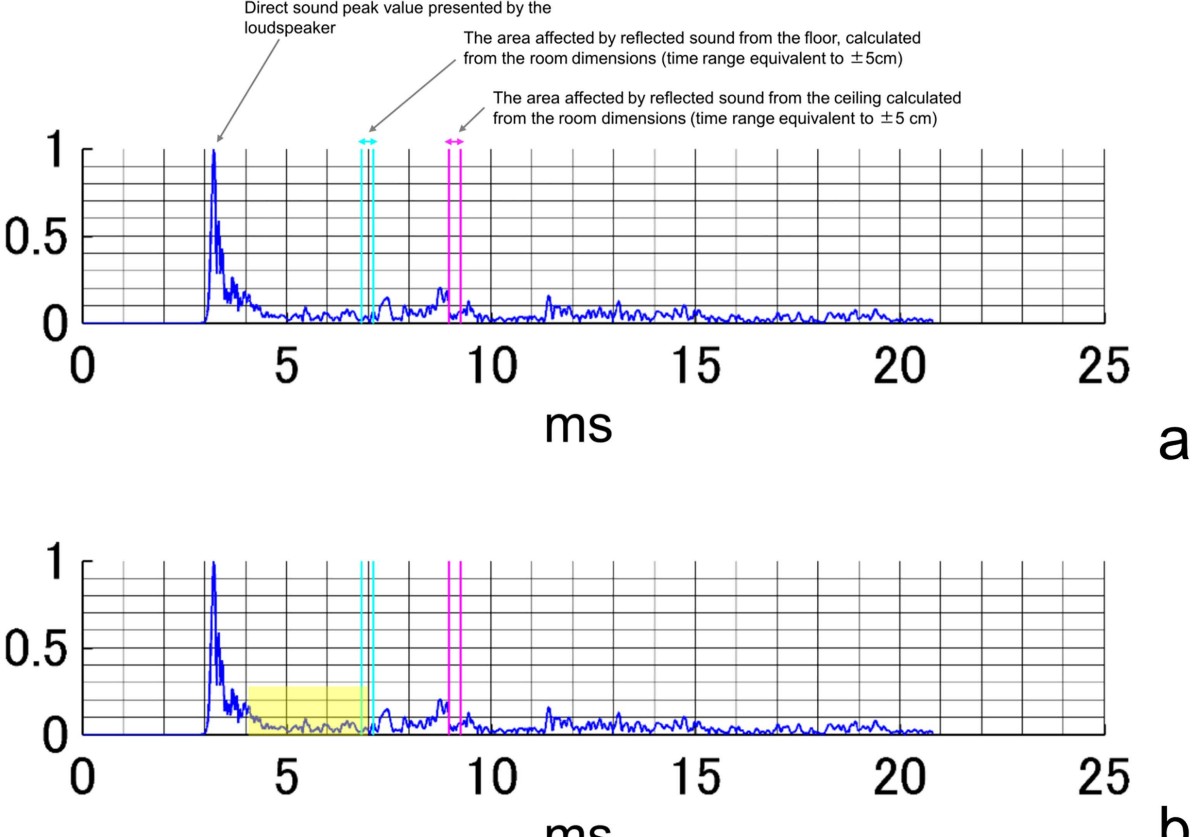

**Fig 2. An example of the results of a reflection measurement. (a)** The maximum amplitude of the presented sound is considered 1, and the vertical axis of the graph indicates the amplitude. The horizontal axis indicates elapsed time (ms). **(b)** A sample waveform segment was extracted using a 4–7 ms time window, illustrating the analysis range used for quantifying the reflected sound envelope and peak value.

To analyze the correlation between the mean number of correct responses and the reflected sound, the acoustic waveforms were clipped with a start position of 3–8 ms and a window width of 1–6 ms, yielding 36 patterns. Fig 2b shows a representative example of a waveform segment extracted using a 4–7 ms window, which was one of the time ranges used to quantify the envelope area and peak amplitude of the reflected sound. The results were divided into four patterns based on the presence or absence of reflected sound effects from the floor and ceiling (i.e., from the floor only, from the ceiling only, and from both the floor and ceiling). Five combinations of patterns were also used for the loudspeakers, with two, four, six, eight, and ten loudspeakers arranged in symmetrical positions. Correlations among the mean number of correct responses, area of the envelope, and peak values of the reflected sound were analyzed for these 720 patterns. Furthermore, correlations between the left and right differences in the number of correct responses, the area of the envelope, and the peak values of the reflected sound were analyzed for 480 patterns. The same conditions were used for the presence or absence of reflected sound effects from the ceiling and floor. Other analysis conditions were as follows: acoustic waveforms were clipped with a start position of 3–7 ms and a window width of 1–6 ms (yielding 30 patterns), and four combinations of patterns were used for the loudspeaker, with two, four, six, and eight loudspeakers arranged in symmetrical positions.

**Sound-localization test**

The participant was seated in front of nine loudspeakers (6301NX, Fostex, Foster, Tokyo, Japan) placed at 22.5° in a semi-circle with a 1 m radius (−90° to 90° azimuth) (Fig 1). The height of the participant's head was adjusted to the height of the loudspeaker, which was placed 1 m above the floor.

The following instructions were provided before the test, to prevent head movement by the participant: (1) Confirm the positions and numbers of speakers in advance. (2) Keep the eyes fixed on the speaker directly in front of you. (3) Keep the head and neck fixed in the direction of the speaker directly in front of you. (4) Do not move your head or neck during the test.

The test was performed as described previously [8,9]. The stimulus was a 1-s Comité Consultatif International Téléphonique et Télégraphique (CCITT) noise or a low-pass CCITT noise burst with a 100-ms rise/fall time in both cases (Fig 3a). The low-pass CCITT noise (Fig 3b) was created using the Audacity software, which involved filtering the CCITT noise to attenuate it by 48 dB per octave above 1500 Hz [10,11]. Audacity is a freely available open-source audio recording and editing software that functions across multiple platforms. The CCITT noise and low-pass CCITT noise conditions were tested in separate sessions. Participants were not exposed to randomized interleaving of different noise types within the same trial block, thereby avoiding uncertainty in spectral content during testing. The stimulus levels were randomly set (50, 55, and 60 dB SPL). A total of 27 unique stimuli were created by combining three stimulus levels (50, 55, and 60 dB SPL) with nine loudspeaker positions. Each condition was presented twice in succession as a single trial, and participants responded once per trial. In total, 27 trials were conducted in random order. The tests were presented as source-identification tasks [12]. The loudspeakers were consecutively numbered from 1 to 9 (−90° to 90° azimuth), and the participant had to identify the loudspeaker considered to be the source of the stimulus by these numbers. During the testing, no feedback was provided. The software for the localization test system was run on a laptop computer and a tablet device and was used for stimulus presentation, data collection, data analysis, and receiving the participants' responses [8,9].

The localization accuracy was quantified using the mean deviation score (d) and the bias score (b). The mean deviation score (d) indicates the deviation between the judged azimuth and the sound presentation azimuth with and without bias adjustment, where the bias is the localization error, which is constant across the loudspeakers. Root mean square (RMS) is a statistical metric used to evaluate the precision or variation in directional perception. It is defined as the square root of the mean of the squared differences between the perceived and actual directions. Both groups (control and UHL) underwent sound-localization testing under the same acoustic conditions and procedures. The results were analyzed using the same statistical model and predictor variables to ensure consistency and comparability between groups.

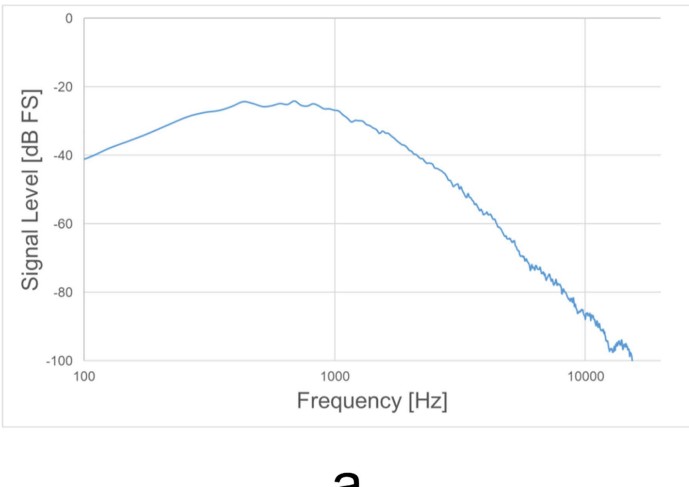
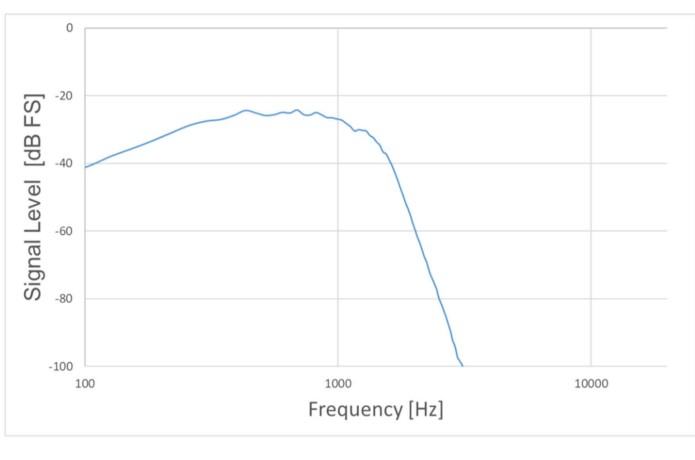

<div align="center">a b</div>

**Fig 3. Frequency response of Comité Consultatif International Téléphonique et Télégraphique (CCITT) noise (a) and low-pass CCITT noise (b).**

## Statistical analyses

IBM SPSS Statistics software (version 29; IBM Corp., Armonk, NY, USA) was used for the statistical analyses. The normality distribution of the analyzed data was evaluated using a Q-Q plot. The observed data points aligned closely with the theoretical reference line, indicating that the data conformed to a normal distribution, confirming the appropriateness of statistical methods that assumed normality in subsequent analyses. Pearson's correlation analyses and *t*-tests were performed.

The significance level was set at 5%. As this study was exploratory in nature, corrections for multiple comparisons were not applied.

## Ethics review

This study was approved by the Ethics Review Committee of Hiroshima University, Hiroshima, Japan (No. E2022-0269). The Central Ethics Review Committee of Hiroshima University approved the study at other sites. The patients/participants provided their written informed consent to participate in this study.

## Results

### Control study (measurement of reflected sound)

The mean hearing level of the control participants in whom pure-tone audiometry was performed was 9.5 dB HL. The number of correct responses per loudspeaker (maximum 21: one response per trial × three sound pressure × seven cases at each facility) for the sound-localization test is shown for each of the 11 facilities (Fig 4a). The number of correct responses was highest at the 0-degree loudspeaker and gradually decreased as the stimulus location moved toward ±90 degrees. However, the downward trend in the number of correct answers varied from facility to facility. Additionally, Fig 4b shows the difference in the number of correct responses for loudspeakers in symmetrical positions for each facility. The average number of correct responses for all loudspeakers and the two channels with the lowest number of correct responses (ch 1 and ch 9) were significantly correlated (r = 0.9363, *p* < .001) (S2 and S3 Figs).

The peak value and area of the reflected sound envelope correlated negatively with the mean number of correct responses. In the analysis of the mean number of correct responses and the area of the envelope, all values showed a negative correlation after 4 ms, and when the negative correlation coefficients were ranked in order from largest to

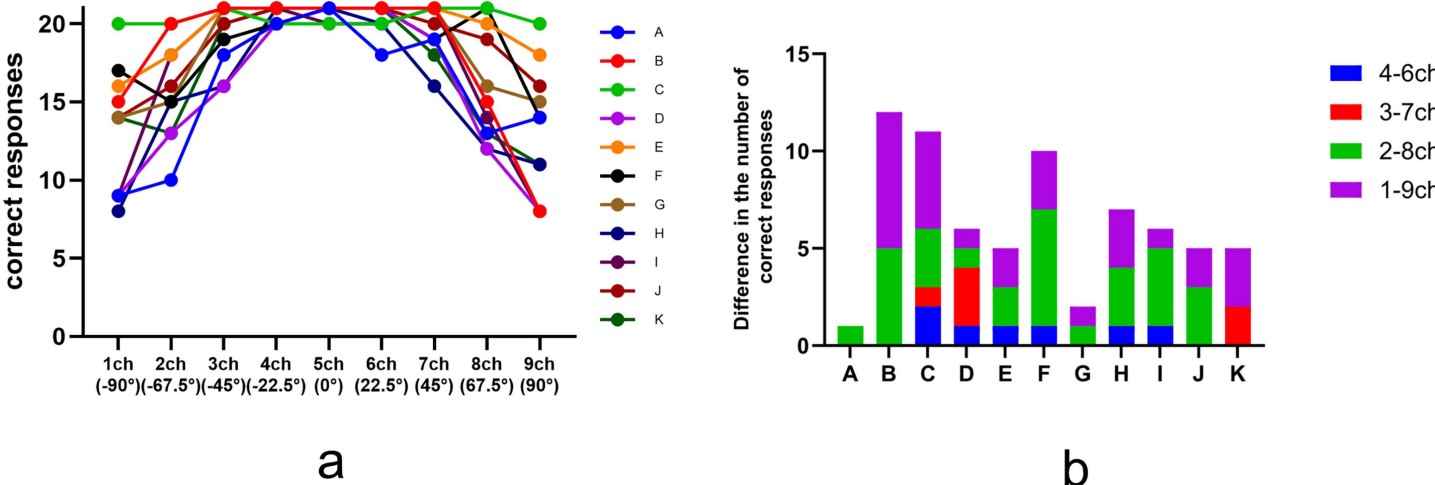

**Fig 4. The number of correct responses for each loudspeaker in the control participants (a) and the difference between the number of correct responses for the loudspeaker on symmetrical position (b) for each facility.**

smallest, seven of the bottom 10 patterns were in the range of 4–7 ms after the influence of the floor and ceiling was removed (r = −0.535 to −0.555). A negative correlation was also observed between the peak values of the reflected sound and the average number of correct responses after 4 ms. When the negative correlation coefficients were arranged from largest to smallest, nine of the bottom 10 patterns were found in the range of 4–7 ms without being related to the reflection influence of the floor and ceiling (r = −0.437 to −0.497) (S4 and S5 Figs).

There was a positive correlation between the envelope area and the difference in the number of correct responses between the left and right sides. In particular, the difference in the number of correct responses between ch 3 and ch 7— corresponding to symmetrically positioned speakers at −45° and 45°, respectively—showed a strong correlation of r > 0.6 in 79–88% of the items in the envelope area and the 4–10-ms extraction section, without being related to the reflection influence of the floor and ceiling. Additionally, in terms of the relationship between the peak value of the acoustic wave-form and the difference in the number of correct answers between the left and right sides, 10 items in ch 3–7 (−45° and 45°), 10 items in ch 2–8 (−67.5° and 67.5°), and two items in ch 1–9 (−90° and 90°) yielded a correlation coefficient of r > 0.6 in the 4–9-ms extraction section, regardless of the reflected sound from the floor and ceiling. A strong correlation was observed between the area and the peak value of the reflected sound in the range of 4–7 ms for sounds presented by ch 1 (−90°), 2 (−67.5°), 8 (67.5°), and 9 (90°) speakers (r = 0.636–0.947). No significant correlation with room size, reverberation time, or background noise was found with the envelope area and peak values of the acoustic waveforms.

## Control study (sound-localization test)

The RMS error for all loudspeakers in the control group with normal CCITT noise was 2.0° ± 4.8°, while the RMS error for low-pass CCITT noise was 2.2° ± 5.2°, which was not significantly different for each loudspeaker (Fig 5a and S1 Dataset). The minimum and maximum responses of the participants were analyzed for each of the nine loudspeakers. The minimum RMS value was zero for all loudspeakers, whereas the maximum RMS value ranged from 9.19 to 27.56.

## UHL study (sound-localization test)

In the UHL group, sudden sensorineural hearing loss was the most common cause of hearing loss (19 cases), followed by congenital hearing loss (5 cases), postoperative ear hearing loss (3 cases), and mumps-related hearing loss (2 cases).

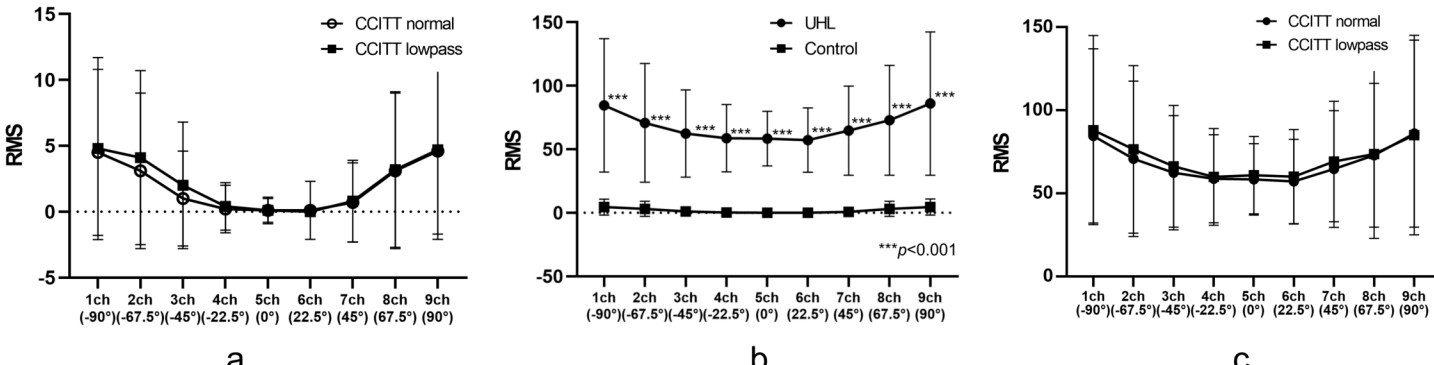

**Fig 5. Comparison of root mean square (RMS) values at each loudspeaker with normal Comité Consultatif International Téléphonique et Télé-graphique (CCITT) noise and low-pass CCITT noise in the control group (a).** Comparison of RMS values at each loudspeaker between unilateral hearing loss and control groups (b) and comparison of RMS values at each loudspeaker in cases of unilateral hearing loss with normal CCITT noise and low-pass CCITT noise (c). In cases of unilateral hearing loss, the side of the worse-hearing ear was considered to be ch 9. If the left ear was the worse-hearing ear, the results were analyzed symmetrically.

The duration from the onset of hearing loss to testing in patients with UHL, excluding those with congenital hearing loss, ranged from 4 months to 74 years, with an average of 17.8 years. The average hearing level of the better-hearing ear in the case of UHL was 21.6 dB, and the average hearing level of the worse-hearing ear was 98.9 dB. The average hearing level at 8 kHz in the worse-hearing ear was 93.9 dB (range: 55–115 dB), while that in the better-hearing ear was 38.4 dB (range: 0–100 dB). The RMS error for all loudspeakers for the CCITT noise in the UHL cases was 68.4°±40.7° (Fig 5b and S2 Dataset). The minimum and maximum responses of the participants were analyzed for each of the nine loud-speakers. The minimum RMS values in the UHL case ranged from 0 to 12.99, whereas the maximum values ranged from 90 to 180. The RMS error values of the UHL cases were significantly greater than those of the control group for every loudspeaker (p < .0001). Among the 45 cases of UHL, 25 were tested using low-pass CCITT noise; however, the results did not show any significant difference when compared to those obtained with standard CCITT noise (Fig 5c). In cases of UHL, the side of the worse-hearing ear was considered to be ch 9, and if the left ear was the worse-hearing ear side, the results were analyzed symmetrically. To examine the potential influence of age on sound-localization performance, an analysis of covariance (ANCOVA) was conducted with age as a covariate. The difference in performance between the UHL and control groups remained significant across all loudspeaker positions (p < .001).

## Discussion

This study presents a comprehensive effort to standardize the laboratory conditions for sound-localization testing in Japan, which has not been reported previously. These findings underscore the significant impact of early reflected sounds on sound-localization accuracy, even in soundproof environments. We conducted a detailed examination to identify factors influencing test outcomes while performing sound-localization assessments in both control participants and individuals with UHL under identical conditions. We showed that sound-localization accuracy was negatively influenced by early reflections, with reduced accuracy correlating with reflected sound envelope area and peak values within 4–7 ms. Par-ticipants with UHL exhibited significantly greater RMS error in sound localization than did normal-hearing individuals. Asymmetries in the left–right response accuracy correlated positively with the reflected sound characteristics (r > 0.6), while noise type (normal vs. low-pass CCITT) did not significantly impact performance in either those with normal hear-ing or those with UHL. Given the age difference between the UHL and control groups, we considered the possibility that age could have confounded the observed group differences in sound localization performance. However, even after statistically accounting for age, the group difference remained strong. This suggests that the poorer sound localization

performance in the UHL group is unlikely to result from age-related auditory decline, but rather reflects genuine differences related to hearing asymmetry. In our study, the low-pass CCITT noise was filtered at 1500 Hz, which primarily removes higher-frequency spectral cues. However, sound-localization performance both in participants with normal hearing and in those with UHL is thought to rely more heavily on low-frequency ITDs and IPDs in such conditions. Therefore, the removal of high-frequency components may not have significantly disrupted the critical localization cues utilized by participants in either group [2,3].

The experimental setup consisted of nine loudspeakers arranged on a horizontal plane, delivering one second of CCITT noise. Standardized equipment was deployed across 11 facilities with diverse laboratory conditions, enabling a reliable cross-comparison of results. Notably, this study highlights that early reflected sounds, delayed by 4–7 ms, overlap with the primary stimulus and markedly affect directional perception. This association was most prominent in channels 1 (−90°), 2 (−67.5°), 8 (67.5°), and 9 (90°), where early reflections exhibited a strong negative correlation with the localization accuracy.

Various methodologies for sound-localization testing have been reported across different facilities, each characterized by variations in the number and placement of loudspeakers, types of noise employed, and testing room conditions [13,14]. Sound-source localization involves two distinct aspects. The first is called absolute localization, which refers to the ability to determine the absolute position of a sound source in a three-dimensional space. In a typical adult experiment, participants sit in an anechoic chamber and keep their heads fixed to the center of spherically arranged loudspeakers. The sound is played from a randomly selected loudspeaker, and the listener points to or gazes at the loudspeaker that appears to be the source of the sound. The second aspect is relative localization, which refers to the detection of absolute misalignment of the sound source. Relative localization is quantified using the minimum audible angle, which is defined as the smallest detectable shift in the angular position of the sound source [15,16]. For routine clinical applications, sound-localization testing must produce consistent results across facilities, while remaining straightforward to implement. Vertical sound localization, although valuable, often presents practical challenges due to the height requirements and spatial constraints in testing environments. Consequently, we propose focusing on horizontal-plane testing within a 180° range, as this aligns with both the practical limitations and the typical spatial orientation of auditory stimuli in real-world scenarios.

The sound-localization test results for control cases revealed frequent false responses to loudspeaker sounds presented near −90° or 90°. Some facilities demonstrated more than 10 discrepancies (out of 21 responses) in the number of correct responses for loudspeakers in symmetrical positions, primarily distributed between channels 1 (−90°) and 9 (90°) and channels 2 (−67.5°) and 8 (67.5°). Yost reported that when young adults with normal hearing were tasked with localizing narrowband noise at 1/10 octave or 2-octave band noise (centered at 250, 2000, and 4000 Hz, positioned between −75° and +75°), neither the sound pressure level (tested at four steps between 20 and 80 dBA), nor the duration of the noise bursts (25, 150, or 450 ms) affected localization accuracy [17]. Although their study did not include positions at −90° and +90°, it demonstrated an increase in the RMS error as the sound source deviated farther from the frontal position, which was consistent with our findings.

Our analyses revealed that the acoustic properties of early reflections—specifically their envelope area and peak value—were associated with the number of correct responses and asymmetries in left–right localization. Specifically, the envelope area of the reflected sounds correlated negatively with the number of correct responses and correlated positively with asymmetries in the left–right response accuracy. It is well established that typical listening environments contain noise, which can mask auditory signals and affect both binaural cues and monaural spectral cues. These effects often result in reduced sound localization and impaired speech comprehension, even in listeners with normal hearing [10]. Research on listeners with normal hearing has demonstrated that the localization accuracy decreases as the signal-to-noise ratio decreases [18]. Temporal analysis indicated that the peak of the presented sound occurred at approximately 3.2 ms and correlated with the envelope area and peak value of reflected sounds within a 4–7-ms window. In reverberant environments, listeners encounter direct and reflected sounds from the source and from room surfaces,

respectively. While the auditory system prioritizes direct sounds during brief stimuli (e.g., clicks), a phenomenon known as the "precedence effect" or echo suppression [19,20], our findings suggested that, in smaller testing environments, early reflections may have occurred with very short delays, thereby reducing the effectiveness of the precedence effect and potentially degrading sound-localization performance. Notably, variables such as room size, reverberation time, and background noise were not significantly correlated with localization accuracy. This suggests that early reflections, rather than the broader acoustic characteristics of the room, are more critical in affecting sound-localization outcomes. These findings underscore the importance of controlling the early reflections in relatively small testing environments. Employing sound-absorbing materials to mitigate reflected sounds can reduce incorrect responses and asymmetries between the left and right sides, thereby enhancing the reliability of the sound-localization tests.

The type of noise (normal vs. low-pass CCITT) did not significantly affect the RMS error in either the control or UHL groups, indicating that sound-localization accuracy is relatively robust to variations in noise bandwidth under horizontal-plane hearing conditions, even though it was hypothesized that localization performance would deteriorate more under low-pass noise because of the lack of spectral cues. In this study, the average hearing level at 8 kHz in the better-hearing ear of the UHL group was relatively well-preserved. When hearing loss in the better-hearing ear at 8 kHz exceeded approximately 30 dB HL, the azimuthal gains were consistently small. Good high-frequency hearing in the functional ear appears to be a critical factor in achieving adequate sound-localization performance [11]. On the other hand, the UHL cases in this study included many individuals with a long disease duration, suggesting the possibility that compensatory mechanisms for sound localization may have been activated. A study examining sound localization in UHL cases with the same hearing levels as those in this study reported that localization performance on the impaired-ear side in UHL patients improved with a longer disease duration or earlier onset of UHL [21]. In this study, one of the factors contributing to the lack of statistical significance of differences between normal and low-pass CCITT was the small sample size of UHL cases. Further investigations involving larger numbers of patients are thus necessary.

In this study, we aimed to minimize interfacility discrepancies when conducting sound-localization tests at different institutions. This would allow universally applicable and comparable testing in routine clinical practice. These findings suggest that reliable testing with minimal errors can be achieved by reducing the initial reflections, even in rooms of varying sizes or with different wall materials, for example, by installing sufficient sound-absorbing materials on immovable structures such as walls and pillars.

A major limitation of this study was the small number of cases included per facility. In the future, we aim to conduct further investigations with a larger number of cases, including tests conducted using hearing-assistive devices. Further accumulation of cases of UHL is also necessary to clarify differences in sound-localization performance between cases of congenital and acquired hearing loss.

## Conclusions

Sound-localization tests conducted in soundproof rooms used in daily clinical practice suggested that early reflections may influence localization performance and that reducing these reflections—such as through the use of sound-absorbing materials—could help minimize inter-facility variability. In addition, in cases of UHL, sound-localization performance was very poor, not only on the affected side but also on the unaffected side. These findings provide insight into the impact of reflected sounds on spatial hearing and highlight the challenges faced by individuals with UHL in sound-localization tasks. Further research could explore interventions to improve localization abilities, particularly in individuals with hearing impairments.

## Supporting information

**S1 Dataset. Raw data from control participants in the sound-localization test.**
(XLSX)

**S2 Dataset. Raw data from UHL participants in the sound-localization test.**
(XLSX)

**S1 Fig. Graphical abstract of study aims, methods, and findings.**
(TIF)

**S2 Fig. Q-Q plots of the peak value and the mean number of correct responses.**
(TIFF)

**S3 Fig. Q-Q plots of the area of the reflected sound envelope and the mean number of correct responses.**
(TIFF)

**S4 Fig. Q-Q plots of the peak value and the difference in the number of correct responses.**
(TIFF)

**S5 Fig. Q-Q plots of the area of the reflected sound envelope and the difference in the number of correct responses.**
(TIFF)

## Author contributions

**Conceptualization:** Tadao Yoshida.

**Data curation:** Tadao Yoshida, Takashi Ishino.

**Formal analysis:** Tadao Yoshida, Takashi Ishino, Kazuki Nishida.

**Funding acquisition:** Tadao Yoshida.

**Investigation:** Tadao Yoshida.

**Methodology:** Tadao Yoshida.

**Project administration:** Tadao Yoshida.

**Resources:** Tadao Yoshida.

**Software:** Takeshi Nakaichi.

**Validation:** Tadao Yoshida.

**Visualization:** Tadao Yoshida.

**Writing – original draft:** Tadao Yoshida.

**Writing – review & editing:** Tadao Yoshida, Takashi Ishino, Satoshi Iwasaki, Naoki Oishi, Yusuke Matsuda, Tetsuya Tono, Kazuma Sugahara, Hiroshi Yamazaki, Sumito Jitsukawa, Hiroshi Nakanishi, Ryosuke Kitoh, Takashi Sato, Masumi Kobayashi.

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
