## [Decision Letter · Decision Letter 0]

6 May 2025

PONE-D-25-14950Establishing standardized conditions for clinically available sound-localization tests: A multicenter approachPLOS ONE

Dear Dr. Yoshida,

Thank you for submitting your manuscript to PLOS ONE. After careful consideration, we feel that it has merit but does not fully meet PLOS ONE’s publication criteria as it currently stands. Therefore, we invite you to submit a revised version of the manuscript that addresses the points raised during the review process.

Your manuscript has been reviewed by two experts in the topic. As you will see below, both reviewers found your study relevant and well-motivated, but they also had a number of concerns, including missing methodological details that would be required to replicate this study, and an insufficient contextualization of this work in the literature on spatial hearing. Both reviewers make very clear and detailed recommendations which I invite you to address in a revision of the paper. At this stage, I cannot guarantee publication, and the quality of your revisions would need to be assessed again, if possible by the same reviewers.

We look forward to receiving your revised manuscript.

Kind regards,

Patrick Bruns

Academic Editor

PLOS ONE

[I have read the journal's policy and the authors of this manuscript have the following competing interests: This was a joint research project with RION Co., Ltd., which assisted purely in measuring the reflected sound. RION Co., Ltd. was not involved in any other aspect of this research, including the planning and implementation, data collection, analysis and interpretation, decision to publish, preparation, review, or approval of the paper.].

3. In the online submission form, you indicated that [The data underlying the results presented in the study are available from Tadao Yoshida.].

Additional Editor Comments (if provided):

Reviewers' comments:

Reviewer's Responses to Questions

**Comments to the Author**

1. Is the manuscript technically sound, and do the data support the conclusions?

Reviewer #1: Yes

Reviewer #2: Partly

2. Has the statistical analysis been performed appropriately and rigorously? 

Reviewer #1: Yes

Reviewer #2: Yes

3. Have the authors made all data underlying the findings in their manuscript fully available?

Reviewer #1: No

Reviewer #2: No

4. Is the manuscript presented in an intelligible fashion and written in standard English?

Reviewer #1: No

Reviewer #2: No

5. Review Comments to the Author

Reviewer #1: The study explored localization abilities of sounds in the azimuthal dimension in order to establish standardized acoustic conditions for sound localization assessments in unilateral hearing loss and control individuals.

The multicenter approach is a plus-value. The method is clear. Establishing standardized tests is a key objective. I think the Results section could be improved, as well as the discussion. I think there is a lack of references in the introduction (eg. my comment 5).

I noticed some mistakes (eg. sentences written twice). The details are reported below in my comments.

It seems that the data availability statement does not fit with the requirements of Plos ONE. They state “No – some restrictions will apply” without explaining the exceptional situation as asked in the guidelines. I do not think this is a reason to reject the paper but I think it is necessary to explain why data is available only upon request to the author Tadao Yoshida.

General comments:

1) It is unclear whether the study was financially supported. In the Financial Disclosure you mention that “the author(s) received no specific funding for this work” but in the Acknowledgment section you mention “This research was supported by AMED under Grant Number JP24dk0310128.”. Could you clarify this please?

2) In the figure, I think it would be clearer if you also mention the azimuth position of the loudspeakers 1CH/2CH […]/ 9CH.

3) Please clarify the data availability statement to fit with the requirements of the journal.

Specific comments :

1) Lines 101-105 : You mention ILD and ITD as acoustic cues for spatial hearing. Please mention Interaural Phase Difference (IPD) which is an important cue when ITD and ILD are not sufficiently reliable.

2) Line 110 : If I have well understood what you mean by “frequency intensity”, you refer to the energy in a frequency band. I think it would be clearer if you use the term “energy” or “spectral component” instead of “frequency intensity”.

3) Lines 114- 115: you write “Thus, since these three elements (ITD, ILD, spectral cues), and changes in sound pressure due to the head-shadow effects are important in directional perception […]”. Can you explain why you mention separately ILD and the changes in sound pressure du to the head-shadow effects?

4) Lines 118-120 : You state “In general, sound-localization test results are worse in patients with unilateral hearing loss (UHL) than in healthy participants, and cognition on the side of the hearing-impaired ear is worse than that on the side of the normal-hearing ear. “ . I think there is a lack of references. Please mention references for sound-localization performance lower in patients with UHL than non-impaired. Also I think you could specify what you are referring to when you say “cognition” (for instance, give examples of tasks and references).

5) Lines 131 and 136: The ages of UHL patients and control individuals are different. Did you test the effect of age on performance? Auditory perception capabilities (comprising auditory localization) tend to decrease with age. I think you could mention it in the discussion.

6) Lines 275-276: It seems something is wrong with the sentence “A positive correlation of the envelope area with the differences between the number of correct responses between the left and right sides”. Is there a verb missing ? Please check this

7) Lines 318-379: the following sentence is already written line 299 : “In cases of UHL, the side of the worse-hearing ear was considered to be ch 9, and if the left ear was the worse-hearing ear side, the results were analyzed symmetrically.” Please check this

8) In the Result section, I think it is not enough clear whether the first results concern control individuals and UHL patients or just control individuals. Could you clarify this please? Also, I think it is not enough clear whether you tested the same predictors for the control group and the UHL patients. You mention effects for the control group but it is not clear if you this effects are also present in the UHL group. Is there interactions with the group (patients versus controls)?

9) Line 332: Do you have explanations for the non-effect of the noise type on performance ? I think it could be clearer if you mention again the low-pass filter (1500 Hz).

10) Line 339. In the discussion you mention “channels 1, 2, 8 and 9”. I think the discussion would be improved if you mention also the azimuth position in degree. Therefore the reader can easily linked the positions of your loudspeakers with the positions mentioned in the discussion (line 366 for instance).

11) Line 359-360: In the sentence “Some facilities demonstrated more than 10 discrepancies in the number of correct responses for loudspeakers in symmetrical positions”, please mention the total number of responses (eg 10 on how many?).

12) Lines 411-413: I think the readers could be interested in practical examples mentioned here on how to reduce the initial reflections (eg, absorbing materials..).

Reviewer #2: This paper describes a multi-center study to evaluate the effect of room reverberation on sound source localization for adults with vs. without unilateral hearing loss. Broadband and lowpass filtered noise was used. Results replicate two well-known findings: better localization accuracy at midline as compared to off midline, and deficits of localization for patients with unilateral hearing loss. Overall, similar results were obtained with the full-band and low-pass filtered noise. Of greatest interest here, performance differed between test sites. Acoustic analysis of sound recorded at each site indicate that performance is detrimentally affected by early reflections. Modifications of the test environments to reduce those reflections could presumably increase consistency of data cross sites.

The ultimate goal of this work, to configure and implement protocols for clinical assessment of localization, is valuable and motivated well. The research described here seems to be a reasonable first step towards this goal, with the caveat that some parameters are underspecified or non-standard. My primary comments have to do with the presentation of this work. There are some details missing that would be required to replicate this study, and the text does not provide enough information to understand the greater context of this work in the literature on spatial hearing.

I can guess why the stimulus parameters tested here were chosen – e.g., level rove to avoid reliance on a loudness cue and filtering to evaluate the relative roles of ITD and ILD – but I think readers would benefit from an explicit discussion of those parameters, including references to published work documenting those effects in other populations.

If I am following correctly, the full bandwidth and lowpass filtered stimuli were interleaved within a block of trials. If that is the case, then pre-trial uncertainty regarding spectral content could theoretically have interfered with the use of spectral shape as a cue to location. That seems like a point for discussion.

Detailed comments follow.

Line 108: These factors would introduce frequency-specific attenuation, but they wouldn’t increase or decrease the frequency of a sound. This paragraph was challenging for me to understand because the phrase “frequency intensity changes” was initially confusing. I would have found this easier to read if it had instead used the phrase “spectral shaping”.

Line 119: It is not clear how cognition relates here. Should this be “recognition”?

Line 121: Unilateral hearing loss does not always eliminate access to sound in the affected ear. Sometimes it just reduces access to sound or degrades the quality of that sound.

Page 6, middle: It is not clear what units are used for defining pure-tone thresholds. If those values are in dB HL, then the normal-hearing controls don’t really have normal hearing in the traditional sense. Demographic information should also be provided for the NH group.

Page 7, bottom: The position and selection of loudspeakers is not clear at this point in the text. Why was the speaker with the “smallest noise out of the three” the one that was selected?

Line 170: It is not clear what the “reflex within its envelope” means.

Page 8, middle: I got very confused about the various patterns – how they were characterized or used.

Line 214: The causal structure of this sentence does not make sense to me. Unless I am missing something, the number of presentations does not affect the number of different stimuli.

Line 251: Is this right -- the maximum number correct for each speaker and center is 21? I understood that there were 2 repetitions at each of 3 levels for each of 7 listeners (2*3*7 = 42).

Line 253-254: This sentence is difficult to parse. I think you mean that performance was best at 0° and decreased as the stimulus deviated from midline.

Line 287: Consider putting information about the normal control group in a separate paragraph, for clarity.

Line 318-319: It wasn’t clear to me how this numbering scheme affected analysis of results, which were evaluated in degrees (I think).

Line 359: The number of discrepancies is only informative in the context of the total number of trials. Consider reporting the percent.

Line 369-370: I don’t understand how this topic sentence relates to the remainder of the paragraph.

Line 383: It does not seem accurate to say that early reflections compromise the precedence effect. It might be more accurate to say that the precedence effect is stronger for longer delays.

Line 397: I may have missed it, but I don’t think this was discussed in the results.

Line 422: This is likely to be true, but I don’t think it is accurate to say that this study showed effects of using sound-absorbing materials.

Figure 1: This figure is not needed and potentially confusing, since there isn’t really an intervention in the conventional sense.

6. PLOS authors have the option to publish the peer review history of their article (what does this mean? ). If published, this will include your full peer review and any attached files.

**Do you want your identity to be public for this peer review?** For information about this choice, including consent withdrawal, please see our Privacy Policy .

Reviewer #1: **Yes: ** Camille Bordeau

Reviewer #2: No

---

## [Author Response · Author response to Decision Letter 1]

19 May 2025

Review Comments to the Author

Reviewer #1: The study explored localization abilities of sounds in the azimuthal dimension in order to establish standardized acoustic conditions for sound localization assessments in unilateral hearing loss and control individuals.

The multicenter approach is a plus-value. The method is clear. Establishing standardized tests is a key objective. I think the Results section could be improved, as well as the discussion. I think there is a lack of references in the introduction (eg. my comment 5).

I noticed some mistakes (eg. sentences written twice). The details are reported below in my comments.

It seems that the data availability statement does not fit with the requirements of Plos ONE. They state “No – some restrictions will apply” without explaining the exceptional situation as asked in the guidelines. I do not think this is a reason to reject the paper but I think it is necessary to explain why data is available only upon request to the author Tadao Yoshida.

General comments:

1) It is unclear whether the study was financially supported. In the Financial Disclosure you mention that “the author(s) received no specific funding for this work” but in the Acknowledgment section you mention “This research was supported by AMED under Grant Number JP24dk0310128.”. Could you clarify this please?

Thank you for pointing out this discrepancy regarding funding. We apologize for the lack of clarity.

To clarify, this study was indeed supported by the Japan Agency for Medical Research and Development (AMED) under Grant Number JP24dk0310128. The original statement in the Financial Disclosure section, “the author(s) received no specific funding for this work,” was incorrect and has been deleted.

We have updated the Financial Disclosure section to reflect this as follows:

Revised Financial Disclosure:

"This study was supported by the Japan Agency for Medical Research and Development (AMED) under Grant Number JP24dk0310128."

2) In the figure, I think it would be clearer if you also mention the azimuth position of the loudspeakers 1CH/2CH […]/ 9CH.

We have added the azimuth positions of the loudspeakers (ch 1/ch 2 […]/ch 9) to Figures 1, 4, and 5 to enhance clarity and spatial understanding for the readers. Please note that the figure numbers have been updated based on the suggestions from Reviewer 2, and the figures referenced in the revised manuscript may differ from the original numbering.

3) Please clarify the data availability statement to fit with the requirements of the journal.

Thank you for pointing this out. We have revised the Data Availability Statement to align with the journal’s requirements. Additionally, we have uploaded the relevant dataset as a supplemental file for transparency and accessibility.

Specific comments :

1) Lines 101-105 : You mention ILD and ITD as acoustic cues for spatial hearing. Please mention Interaural Phase Difference (IPD) which is an important cue when ITD and ILD are not sufficiently reliable.

The interaural phase difference (IPD) is an important auditory spatial cue that becomes particularly useful when interaural time differences (ITD) and interaural level differences (ILD) are not sufficiently reliable. The IPD refers to the difference in the phase of a sound wave arriving at each ear, and it is especially effective at lower frequencies where phase information is more discernible and ITD and ILD cues can be ambiguous or diminished [2,3] (lines 81–85).

[2] Blauert, J. Spatial hearing: The psychophysics of human sound localization. Revised ed. Cambridge, MA: MIT; 1996.

[3] Zirn S, Arndt S, Aschendorff A, Laszig R, Wesarg T. Perception of interaural phase differences with envelope and fine structure coding strategies in bilateral cochlear implant users. Trends Hear. 2016;20: 233121651666560.

2) Line 110 : If I have well understood what you mean by “frequency intensity”, you refer to the energy in a frequency band. I think it would be clearer if you use the term “energy” or “spectral component” instead of “frequency intensity”.

Thank you for your helpful comment. You are correct in your interpretation — by “frequency intensity,” we intended to refer to the energy within a given frequency band. To improve clarity and accuracy, we have replaced the term “frequency intensity” with “spectral component” throughout the manuscript (lines 89, 90).

3) Lines 114- 115: you write “Thus, since these three elements (ITD, ILD, spectral cues), and changes in sound pressure due to the head-shadow effects are important in directional perception […]”. Can you explain why you mention separately ILD and the changes in sound pressure du to the head-shadow effects?

We agree that mentioning both ILD and the changes in sound pressure due to the head-shadow effect may have been redundant, as ILD largely reflects the acoustic attenuation caused by the head shadow. To clarify this point and to improve the scientific accuracy, we have revised the sentence as follows:

“Thus, since these four elements (ITD, ILD, IPD, and spectral cues) are important in directional perception, a test method that reflects these detection capabilities is needed for accurate sound-localization testing” (lines 94-96).

This revision eliminates the redundant mention of head-shadow effects and incorporates IPD, which plays an important role when ITD and ILD are less reliable, particularly at low frequencies or in complex acoustic environments.

4) Lines 118-120 : You state “In general, sound-localization test results are worse in patients with unilateral hearing loss (UHL) than in healthy participants, and cognition on the side of the hearing-impaired ear is worse than that on the side of the normal-hearing ear. “ . I think there is a lack of references. Please mention references for sound-localization performance lower in patients with UHL than non-impaired. Also I think you could specify what you are referring to when you say “cognition” (for instance, give examples of tasks and references).

As suggested, we have added appropriate references to support the statement that sound-localization performance is poorer in individuals with unilateral hearing loss (UHL) compared to that in normal-hearing individuals (lines 99–102). The following references have been cited in the revised manuscript:

[6] Parisa A, Reza NA, Jalal SS, Mohammad K, Homa ZK. Horizontal localization in simulated unilateral hearing loss. J Audiol Otol. 2017;22: 39–44.

[7] Nelson E, Reeder RM, Holden LK, Firszt JB. Front- and rear-facing horizontal sound localization results in adults with unilateral hearing loss and normal hearing. Hear Res. 2019;372: 3–9.

In addition, we have revised the word “cognition” to “recognition” in this sentence for greater clarity. The revised sentence now reads:

“In general, sound-localization test results are worse in patients with unilateral hearing loss (UHL) than in healthy participants, and recognition on the side of the hearing-impaired ear is worse than that on the side of the normal-hearing ear” (line 98).

5) Lines 131 and 136: The ages of UHL patients and control individuals are different. Did you test the effect of age on performance? Auditory perception capabilities (comprising auditory localization) tend to decrease with age. I think you could mention it in the discussion.

We thank the reviewer for highlighting this point. To evaluate the potential influence of age, we conducted an analysis of covariance (ANCOVA) with age as a covariate. The group difference between UHL and control individuals remained significant (p < .001) for every loudspeaker, indicating that age did not explain the observed differences in performance. We have added this clarification to the Results and Discussion sections (lines 316–320; 335–340).

6) Lines 275-276: It seems something is wrong with the sentence “A positive correlation of the envelope area with the differences between the number of correct responses between the left and right sides”. Is there a verb missing ? Please check this

Thank you for pointing this out. You are correct that the original sentence was unclear due to the absence of a main verb. We have revised the sentence for grammatical clarity and readability. The corrected sentence now reads:

“There was a positive correlation between the envelope area and the difference in the number of correct responses between the left and right sides” (lines 266–267).

7) Lines 318-379: the following sentence is already written line 299 : “In cases of UHL, the side of the worse-hearing ear was considered to be ch 9, and if the left ear was the worse-hearing ear side, the results were analyzed symmetrically.” Please check this

Thank you for your careful reading. While the sentence at line 299 (“In cases of UHL, the side of the worse-hearing ear was considered to be ch 9, and if the left ear was the worse-hearing ear side, the results were analyzed symmetrically.”) appears similar to the sentence in lines 318–379, we would like to clarify that the line 299 sentence appears within the figure legend. As such, it serves to explain the figure independently and does not constitute redundancy within the main text. We believe it is important to retain this information in both locations to ensure clarity for readers who refer to the figure and those who follow the main text.

8) In the Result section, I think it is not enough clear whether the first results concern control individuals and UHL patients or just control individuals. Could you clarify this please? Also, I think it is not enough clear whether you tested the same predictors for the control group and the UHL patients. You mention effects for the control group but it is not clear if you this effects are also present in the UHL group. Is there interactions with the group (patients versus controls)?

We thank the reviewer for this insightful comment. To address the issue, we revised the Results section to clearly distinguish the analyses conducted in each group by adding the following subheadings: Control study (measurement of reflected sound), Control study (sound-localization test), and UHL study (sound-localization test).

Furthermore, we have added the following sentence to the Materials and Methods section under the Sound-localization test heading to clarify the consistency between groups:

“Both groups (control and UHL) underwent sound-localization testing under the same acoustic conditions and procedures. The results were analyzed using the same statistical model and predictor variables to ensure consistency and comparability between groups” (218–221).

We also examined whether there were any interaction effects between the group (control vs. UHL) and the predictors. Both groups showed significant differences in RMS at all loudspeaker positions. Therefore, interaction effects were considered negligible.

9) Line 332: Do you have explanations for the non-effect of the noise type on performance ? I think it could be clearer if you mention again the low-pass filter (1500 Hz).

Thank you for your insightful question. We agree that further clarification is needed regarding the non-significant effect of noise type on performance.

In our study, the low-pass CCITT noise was filtered at 1500 Hz, which primarily removes higher-frequency spectral cues. However, sound-localization performance both in participants with normal hearing and in those with UHL is thought to rely more heavily on low-frequency ITDs and IPDs in such conditions. Therefore, the removal of high-frequency components may not have significantly disrupted the critical localization cues utilized by participants in either group.

We have added this explanation and clarified the 1500 Hz low-pass filtering in the revised manuscript (lines 340–345).

10) Line 339. In the discussion you mention “channels 1, 2, 8 and 9”. I think the discussion would be improved if you mention also the azimuth position in degree. Therefore the reader can easily linked the positions of your loudspeakers with the positions mentioned in the discussion (line 366 for instance).

Thank you for your helpful suggestion. We agree that including the azimuth positions would enhance clarity. We have revised the sentence to read:

“This association was most prominent in channels 1 (-90°), 2 (-67.5°), 8 (67.5°), and 9 (90°), where early reflections exhibited a strong negative correlation with the localization accuracy” (lines 350–352).

11) Line 359-360: In the sentence “Some facilities demonstrated more than 10 discrepancies in the number of correct responses for loudspeakers in symmetrical positions”, please mention the total number of responses (eg 10 on how many?).

Thank you for your comment. To clarify, the "more than 10 discrepancies" refers to the number of mismatched correct responses out of 21 total responses per channel. We have revised the sentence to improve clarity as follows:

“Some facilities demonstrated more than 10 discrepancies (out of 21 responses) in the number of correct responses for loudspeakers in symmetrical positions, primarily distributed between channels 1(-90°) and 9 (90°), and channels 2 (-67.5°) and 8 (67.5°)” (lines 371–374).

12) Lines 411-413: I think the readers could be interested in practical examples mentioned here on how to reduce the initial reflections (eg, absorbing materials..).

Thank you for your valuable suggestion. We agree that providing practical examples for reducing early reflections would enhance the applicability of our findings. Accordingly, we have revised the sentence as follows:

“These findings suggest that reliable testing with minimal errors can be achieved by reducing the initial reflections, even in rooms of varying sizes or with different wall materials, for example, by installing sufficient sound-absorbing materials on immovable structures such as walls or pillars” (lines 424–427).

Reviewer #2: This paper describes a multi-center study to evaluate the effect of room reverberation on sound source localization for adults with vs. without unilateral hearing loss. Broadband and lowpass filtered noise was used. Results replicate two well-known findings: better localization accuracy at midline as compared to off midline, and deficits of localization for patients with unilateral hearing loss. Overall, similar results were obtained with the full-band and low-pass filtered noise. Of greatest interest here, performance differed between test sites. Acoustic analysis of sound recorded at each site indicate that performance is detrimentally affected by early reflections. Modifications of the test environments to reduce those reflections could presumably increase consistency of data cross sites.

The ultimate goal of this work, to configure and implement protocols for clinical assessment of localization, is valuable and motivated well. The research described here seems to be a reasonable first step towards this goal, with the caveat that some parameters are underspecified or non-standard. My primary comments have to do with the presentation of this work. There are some details missing that would be required to replicate this study, and the text does not provide enough information to understand the greater context of this work in the literature on spatial hearing.

I can guess why the stimulus parameters tested here were chosen – e.g., level rove to avoid reliance on a loudness cue and filtering to evaluate the relative roles of ITD and ILD – but I think readers would benefit from an explicit discussion of those parameters, including references to published work documenting those effects in other populations.

If I am following correctly, the full bandwidth and lowpass filtered stimuli were interleaved within a block of trials. If that is the case, then pre-trial uncertainty regarding spectral content could theoretically have interfered with the use of spectral shape as a cue to location. That seems like a point for discussion.

Thank you for you

---

## [Decision Letter · Decision Letter 1]

3 Jun 2025

PONE-D-25-14950R1Establishing standardized conditions for clinically available sound-localization tests: A multicenter approachPLOS ONE

Dear Dr. Yoshida,

Thank you for submitting your manuscript to PLOS ONE. After careful consideration, we feel that it has merit but does not fully meet PLOS ONE’s publication criteria as it currently stands. Therefore, we invite you to submit a revised version of the manuscript that addresses the points raised during the review process.

Your revised manuscript has been reviewed by original Reviewer #1 and myself, as Reviewer #2 was unavailable at this time. I agree with Reviewer #1 that the review comments have been largely addressed. However, as you will see below, Reviewer #1 pointed out a few remaining minor issues which I invite you to address.

We look forward to receiving your revised manuscript.

Kind regards,

Patrick Bruns

Academic Editor

PLOS ONE

Journal Requirements:

Reviewers' comments:

Reviewer's Responses to Questions

**Comments to the Author**

1. If the authors have adequately addressed your comments raised in a previous round of review and you feel that this manuscript is now acceptable for publication, you may indicate that here to bypass the “Comments to the Author” section, enter your conflict of interest statement in the “Confidential to Editor” section, and submit your "Accept" recommendation.

Reviewer #1: (No Response)

2. Is the manuscript technically sound, and do the data support the conclusions?

Reviewer #1: Yes

3. Has the statistical analysis been performed appropriately and rigorously? 

Reviewer #1: Yes

4. Have the authors made all data underlying the findings in their manuscript fully available?

Reviewer #1: Yes

5. Is the manuscript presented in an intelligible fashion and written in standard English?

Reviewer #1: Yes

6. Review Comments to the Author

Reviewer #1: Dear authors,

Thank you for having carefully adressed my comments. I have revised your new submission. I think the manuscript clarity has been improved and the methodology is clearer.

I still would like you to adress two points, linked to my previous 1) and 4) comments.

1) The correction of data availability does not appear in the new submitted manuscript (“the author(s) received no specific funding for this work “ is still mentioned).

4) lines 97-102 : Thank you for the references on poorer localization performance in individuals with UHL, and for the correction of “cognition” into “recognition”. I still do not think there is enough clarity regarding the recognition performance. It would be clearer if you mention the task you are referring to. For instance, if you are referring to speech recognition in noise, please mention a reference for that.

Thank you.

7. PLOS authors have the option to publish the peer review history of their article (what does this mean? ). If published, this will include your full peer review and any attached files.

**Do you want your identity to be public for this peer review?** For information about this choice, including consent withdrawal, please see our Privacy Policy .

Reviewer #1: **Yes: ** Camille Bordeau

---

## [Author Response · Author response to Decision Letter 2]

5 Jun 2025

6. Review Comments to the Author

Reviewer #1: Dear authors,

Thank you for having carefully adressed my comments. I have revised your new submission. I think the manuscript clarity has been improved and the methodology is clearer.

I still would like you to adress two points, linked to my previous 1) and 4) comments.

1) The correction of data availability does not appear in the new submitted manuscript (“the author(s) received no specific funding for this work “ is still mentioned).

Thank you for your comment.

Regarding the Data Availability statement, we confirm that the following sentence has already been included in the revised manuscript:

“All relevant data are within the manuscript and its Supporting Information files.”

However, we acknowledge that the Funding statement was not appropriately updated in the previous version. We have now corrected it in the manuscript to accurately reflect the funding status. The updated sentence reads:

“This study was supported by the Japan Agency for Medical Research and Development (AMED) under Grant Number JP24dk0310128.”

As it appeared that this information was also expected to be included in the cover letter, we have ensured that the updated funding statement is now clearly stated in the resubmitted cover letter.

We appreciate your careful review and apologize for the oversight.

4) lines 97-102 : Thank you for the references on poorer localization performance in individuals with UHL, and for the correction of “cognition” into “recognition”. I still do not think there is enough clarity regarding the recognition performance. It would be clearer if you mention the task you are referring to. For instance, if you are referring to speech recognition in noise, please mention a reference for that.

Thank you for your helpful comment. We understand that the term "recognition" in the original sentence may have caused confusion with speech recognition tasks. To clarify our intended meaning, we have revised the sentence to use the term “directional perception (i.e., the ability to recognize the direction of a sound source)” instead. This change more accurately reflects the context of sound localization performance in individuals with unilateral hearing loss (UHL). The revised sentence now reads:

“In general, sound-localization test results are worse in patients with unilateral hearing loss (UHL) than in healthy participants, and directional perception (i.e., the ability to recognize the direction of a sound source) on the side of the hearing-impaired ear is poorer than that on the side of the normal-hearing ear.”

We believe this clarification resolves the ambiguity and clearly indicates that the statement pertains to localization performance rather than speech recognition ability.

---

## [Editor Report · Decision Letter 2]

10 Jun 2025

Establishing standardized conditions for clinically available sound-localization tests: A multicenter approach

PONE-D-25-14950R2

Dear Dr. Yoshida,

We’re pleased to inform you that your manuscript has been judged scientifically suitable for publication and will be formally accepted for publication once it meets all outstanding technical requirements.

Kind regards,

Patrick Bruns

Academic Editor

PLOS ONE

Additional Editor Comments (optional):

Thank you for addressing the remaining reviewer comments, I am now happy to accept the manuscript for publication.
---

## [Editor Report · Acceptance letter]

PONE-D-25-14950R2

PLOS ONE

Dear Dr. Yoshida,

I'm pleased to inform you that your manuscript has been deemed suitable for publication in PLOS ONE. Congratulations! Your manuscript is now being handed over to our production team.

Kind regards,

on behalf of

Dr. Patrick Bruns

Academic Editor

PLOS ONE